# NK cells modulate in vivo control of SARS-CoV-2 replication and suppression of lung damage

**Harikrishnan Balachandran[1], Kyle Kroll[1], Karen Terry[1], Cordelia Manickam[1], Rhianna Jones[1], Griffin Woolley[1], Tammy Hayes[2], Amanda J. Martinot[2], Ankur Sharma[3], Mark Lewis[3], Stephanie Jost[1], R. Keith Reeves**(ID)[1,4]*

1 Division of Innate and Comparative Immunology, Center for Human Systems Immunology, Duke University School of Medicine, Durham, North Carolina, United States of America, 2 Department of Infectious Diseases and Global Health, Cummings School of Veterinary Medicine at Tufts University, North Grafton, Massachusetts, United States of America, 3 BIOQUAL, Inc., Rockville, Maryland, United States of America, 4 Department of Surgery, Duke University School of Medicine, Durham, North Carolina, United States of America

* keith.reeves@duke.edu

**Data Availability Statement:** All relevant data are included in the manuscript and supplementary material.

## Abstract

Natural killer (NK) cells play a critical role in virus control. However, it has remained largely unclear whether NK cell mobilization in SARS-CoV-2 infections is beneficial or pathologic. To address this deficit, we employed a validated experimental NK cell depletion non-human primate (NHP) model with SARS-CoV-2 Delta variant B.1.617.2 challenge. Viral loads (VL), NK cell numbers, activation, proliferation, and functional measures were evaluated in blood and tissues. In non-depleted (control) animals, infection rapidly induced NK cell expansion, activation, and increased tissue trafficking associated with VL. Strikingly, we report that experimental NK cell depletion leads to higher VL, longer duration of viral shedding, significantly increased levels of pro-inflammatory cytokines in the lungs, and overt lung damage. Overall, we find the first significant and conclusive evidence for NK cell-mediated control of SARS-CoV-2 virus replication and disease pathology. These data indicate that adjunct therapies for infection could largely benefit from NK cell-targeted approaches.

## Author summary

Natural killer (NK) cells play a critically understudied role in controlling SARS-CoV-2 viral replication, clearance, and disease sequelae. In this manuscript, we investigated the protective role of NK cells in acute infection using a well-established NK cell depletion strategy in cynomolgus macaques (CM) and a SARS-CoV-2 Delta variant infection model. Circulating NK cells exhibited an increased proliferative and activated phenotype following infection, concomitant with peak NK cell expansion at 10 days post-infection (DPI). Importantly, following experimental NK cell depletion, CM exhibited increased viral shedding and delayed viral clearance compared to controls. NK cell-depleted animals also exhibited significantly increased lung pathology and Luminex cytokine analyses of broncho-alveolar lavage (BAL) fluid showed a 5-fold increase in interferon-alpha (IFNα) compared to controls during peak infection. Collectively, our findings suggest that NK

**Funding:** This work was supported by the National Institute of Diabetes and Digestive and Kidney Diseases (R01DK130472, to RKR), National Institute of Child Health and Human Development (R01HD103721, to RKR), and National Institute of Allergy and Infectious Diseases (R01AI158516, to RKR). The funders had no role in the study design, data collection and analysis, decision to publish, or preparation of the manuscript.

**Competing interests:** The authors have declared that no competing interests exist.

cells play a crucial role in controlling SARS-CoV-2 replication and reducing lung damage. These results underscore the potential of NK cell-based vaccines and therapies for COVID-19 and other infectious diseases, warranting further investigation in this area.

## Introduction

The novel severe acute respiratory syndrome coronavirus 2 (SARS-CoV-2) is the etiologic agent of coronavirus disease 2019 (COVID-19), a disease which spread rapidly worldwide after its origin in late 2019 and led to an unprecedented global public health crisis [1–3]. Treatment options and vaccines against SARS-CoV-2 were very limited during the initial phases of the pandemic, and their efficacy since development has been challenged by the emergence of variants [4–6]. Despite these shortcomings, recent studies have demonstrated that host immune responses at all levels of the immune system play crucial roles in the containment of SARS-CoV-2 infection [5,7,8], and such discoveries will continue to be vital for the development of effective therapeutic strategies.

Classically, natural killer (NK) cells are viewed as nonspecific effector cells of the innate immune system that play critical roles in defense against viral infections [9,10]. The virus-infected cells upregulate self-encoded molecules induced by the infection and/or cellular stress response (MHC class I-related chains (MIC) A and B and members of the UL16-binding protein (ULBP)) as well as pathogen-derived molecules. NK cells mediate the killing via the engagement of these molecules with the natural cytotoxicity receptors (NKp30, NKp44, and NKp46) and C-type lectin-like receptors (NKG2D and NKp80) on their surface [11–15]. Viral infection can also induce the expression of death receptors on infected cells, which leads to NK cell-mediated cytolysis via the engagement of the Fas ligand (FasL) and tumor necrosis factor-related apoptosis-inducing ligand (TRAIL) [16–18]. Additionally, cytotoxic NK cells facilitate viral clearance directly through caspase-mediated pathways, achieved by the release of stored perforin and granzyme cytolytic granules, triggering apoptosis of the infected cell [17,19]. Indirectly, NK cells release a wide range of proinflammatory cytokines with antiviral activity and can mediate antibody-mediated cellular cytotoxicity (ADCC) [9,20–22]. Our group and others have also previously shown in simian immunodeficiency virus (SIV)-infected non-human primates (NHP) that depleting NK cells leads to increased systemic and tissue viral shedding, indicating that NK cells regulate replication, dissemination, and inflammation during viral infection [23–26].

There has been a plethora of studies in COVID-19 patients and NHP models describing T and B cell responses during acute infection, immediate recovery, and extended convalescence, as well as post-vaccination [27–31]. In a T cell-depleted rhesus macaques (RM) model the delayed but efficient clearance of SARS-CoV-2 infection was orchestrated by virus-neutralizing antibodies [32]. However, despite being an important component of the immune system, less is known about NK cell responses to SARS-CoV-2 infection. During the acute phase of SARS-CoV-2 infection, a severity-dependent lymphopenia affecting T, B, and NK cells has been reported [33–35]. The circulating levels of NK cells in humans declined to 55% during acute infection compared to pre-infection [36], which can be attributed to putative over-recruitment and homing of NK cells into lungs due to SARS-CoV-2-induced cytokine storm [37,38]. Further, transcriptomic analysis of bronchoalveolar lavage (BAL) of COVID-19 infected patients indicated increased levels of the cytokines monocyte chemoattractant protein-1 (MCP-1) and interferon gamma-induced protein 10 (IP-10), which despite being NK cell migration regulators were identified as biomarkers associated with disease severity and

fatality [39–41]. The functional capacity of these migrated NK cells is still being characterized, with mixed reports of enhanced cytotoxicity [35,42] and/or increased production of inflammatory cytokines [43]. Nonetheless, individuals who fail to mount a substantial NK cell-mediated response often have worse disease prognoses and extended viral shedding [44,45]. Taken together, these findings indicate that NK cells may be crucial for the early containment of SARS-CoV-2 and the formation of adaptive responses elicited by infection. However, alternative evidence suggests uncontrolled NK cell responses to infection could contribute to the hyperinflammatory responses and lung damage observed in COVID-19 patients[38,42,46].

In this study we used an established NHP model, which recapitulates viral replication, immune responses, and disease pathology observed in human COVID-19 infection, to evaluate the role of NK cells in primary SARS-CoV-2 infection. To achieve this goal, we investigated the contribution of systemic NK cell mobilization to SARS-CoV-2 pathogenesis and clearance in cynomolgus macaque (CM) models using a well-described method of *in vivo* NK cell depletion specific for NHP [23,47–49]. The results of this innovative study will contribute to our understanding of human immune responses against SARS-CoV-2 and provide the rationale to develop novel immunotherapeutic approaches which target specific NK cell subsets and could substantially strengthen ongoing and alternative efforts for the prevention and treatment of COVID-19.

## Results

### NK cells are mobilized during acute SARS-CoV-2 infection

Twelve CM were selected for this study and divided into two groups. The first group (n = 6) was NK cell-depleted using a sequential intravenous dosage of anti-IL-15 monoclonal antibody (mAb) and the second group (n = 6) received PBS placebo injections (**Fig 1A**). Both NK cell-depleted and non-depleted groups (control) were challenged with SARS-CoV-2 Delta variant B.1.617.2 and then samples were collected over the course of three weeks' infection. NK cell frequencies in circulation for both groups are shown in **Fig 1B** and the baseline (day -14) percentage of NK cells in blood for all animals was 1.59% (mean; range: 0.88%-2.71%). As anticipated, the anti-IL-15 treated animals demonstrated drastically reduced circulating NK cell percentages throughout the duration of the study. We also observed that anti-IL-15-mediated NK cell depletion was highly specific since it did not impact blood bulk T cell frequencies (**Fig 1C**), $CD4^+$ and $CD8^+$ T cells, or B cells (**S1A-S1C Fig**). Despite transient changes in the naïve, central memory, and effector memory $CD4^+$ and $CD8^+$ T cell frequencies, no significant differences between the groups were observed (**S1D–S1I Fig**). This strategy was also highly efficient and durable in tissues as cross-sectional analyses of samples collected at necropsy indicated systemic NK cell ablation (**S2 Fig**). Collectively, these data indicated our NK cell depletion strategy (which has been well-defined by our group and others) [23,47,48], was highly effective and NK cell-specific in this model.

In control animals, longitudinal evaluation suggested significant NK cell mobilization in specific response to infection. At 10 days post-infection (DPI), the control group showed peak NK cell frequencies of 4.09% (mean; range: 3.00%-5.89%) with expansion being significant between pre-challenge and 10DPI and 14DPI (*p*-values < 0.0001 and 0.0004 respectively) (**Fig 1B**). However, as anticipated for an acute viral infection, by 22DPI NK cell levels had returned to baseline. Further, the percentage of peripheral blood NK cells expressing CD69, a marker associated with activation and tissue trafficking/residency [50–52], significantly increased from pre-infection (mean 3.56%; range: 1%-5.93%) to 3DPI (mean 17.96%; range: 3.84%-27.55%) and declined immediately afterwards (adjusted *p*-value < 0.0001) (**Fig 1D**). In peripheral LN there was also a significant increase in NK cells expressing CD69 from the pre-

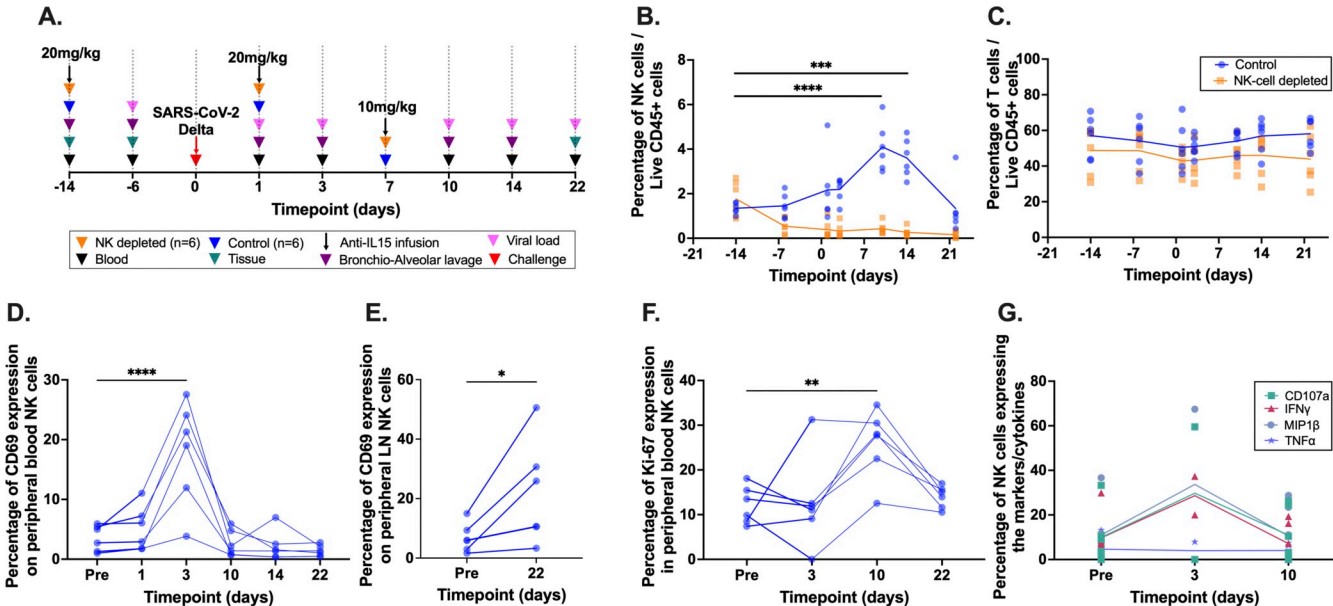

**Fig 1. Study outline and Cellular dynamics in blood following NK cell depletion in acute SARS-CoV-2 infection in cynomolgus macaques (CM). (A)** Study outline; **(B)–(C)** Longitudinal comparison of natural killer (NK) and T cell frequencies in CM peripheral blood mononuclear cells (PBMC) from NK cell-depleted and control groups; **(D)–(E)** Longitudinal expression of activation and trafficking marker CD69 on NK cells in the control group in peripheral blood and LN; **(F)** Longitudinal expression of proliferation marker Ki67 on NK cells in the control group in peripheral blood and **(G)** Longitudinal functional evaluation of NK cells in the control group in peripheral blood. **B, C, D, F and G**–Two-way ANOVA multiple comparison; **E**–Wilcoxon matched-pairs signed rank test Wilcoxon test. * p-value ≤ 0.05; ** adjusted p-value ≤ 0.01; *** adjusted p-value ≤ 0.001; **** adjusted p-value ≤ 0.0001. LN–Lymph Node; CR–Colorectal biopsy.

infection timepoint (mean 6.81% range: 1.64%-14.98%) to 22DPI (mean 21.95% range: 3.32%-50.58%) (*p*-value = 0.0312), suggesting a potential cell redistribution (**Fig 1E**). Post-activation, the expression of Ki-67, a marker of cellular proliferation [53,54], significantly increased from pre-infection (mean 12.14%; range: 7.38–18.1%) to a peak at 10DPI (mean 24.26%; range: 12.54% - 30.48) (**Fig 1F**). Overall, these data indicated NK cell mobilization in number, activation, proliferation, and potential tissue trafficking in response to SARS-CoV-2 challenge.

In humans, peripheral blood NK cells are broadly classified into two subpopulations based on CD56 and CD16 expression as CD56$^{bright}$CD16$^-$ cytokine-secreting and CD56$^{dim}$CD16$^+$ cytotoxic cells. However, in NHP, gating for CD56 and CD16 expression on circulating NKG2A/C$^+$ NK cells delineates three distinct populations: CD56$^+$CD16$^-$ cells which are functionally equivalent to human cytokine-secreting NK cells; CD56$^-$CD16$^+$ cells corresponding to the human cytotoxic NK cells and the CD56$^-$CD16$^-$ (double negative) cells for which a human analog has not yet clearly defined [55,56].

Based on this classification, NK cells in the control group were further divided based on their co-expression profiles of CD56 and CD16. Interestingly, towards the later period of observation, there were significant shifts in three of the four NK cell subsets (**S3 Fig**). There was a significant increase (adjusted *p*-value = 0.042) in the CD56$^+$CD16$^-$ subset frequency from 10DPI to 22DPI (mean 0.69%; range: 0.26%-1.28% and mean 3.05%; range: 6.37%-1.81%, respectively). A similar significant increase (adjusted *p*-value = 0.0005) was also observed in the CD56$^-$CD16$^-$ subset (mean 37.98%; range: 23.36%-50.17% and mean 50.03%; range: 34.90%-68.35% for 10DPI and 22DPI, respectively). Conversely, the CD56$^-$CD16$^+$ subset significantly decreased (adjusted *p*-value = 0.0006) from 10DPI to 22DPI (mean 63.32%; range: 48.54%-75.99% and mean 46.86%; range: 25.22%-61.94%, respectively). Since CD16$^+$

NK cells may represent the most mature population of NK cells, these data suggest immune contraction following viral clearance may be associated with an influx of less differentiated cells.

Finally, NK cells from the control group were evaluated for their functional capacity by measuring degranulation and cytokine secretion using an intracellular cytokine staining assay at longitudinal timepoints. We observed a rapid increase in both cytokine production and surrogate indications of cytotoxicity at 3DPI (macrophage inflammatory protein-1 beta (MIP-1β), interferon-gamma (IFN-γ), tumor necrosis factor-alpha (TNF-α) and CD107a) (**Fig 1G**) followed by a return to baseline levels at 10DPI. Collectively, these data mirror the early activation and mobilization of NK cells followed by rapid contraction.

## NK cells control SARS-CoV-2 virus replication and dissemination

We next evaluated the impact of experimental NK cell depletion on multiple measures of SARS-CoV-2 replication. In BAL fluid and in the pharynx, both subgenomic (Sub G) and total envelop e (Total E) VL peaked between 1DPI and 3DPI (**Fig 2A–2D**). Although acute viral loads were higher in some NK cell-depleted animals, these differences were not significant. However, by 10DPI the differences in BAL Sub G measurements in the NK cell-depleted group (mean 10461; range: 1416–47042, RNA copies/mL) were significantly higher ($p$-value = 0.002) than in the control group (mean 248; range: 0–616, RNA copies/mL) (**Fig 2E**). At the same timepoint, the BAL Total E VL measurements were also significantly higher ($p$-

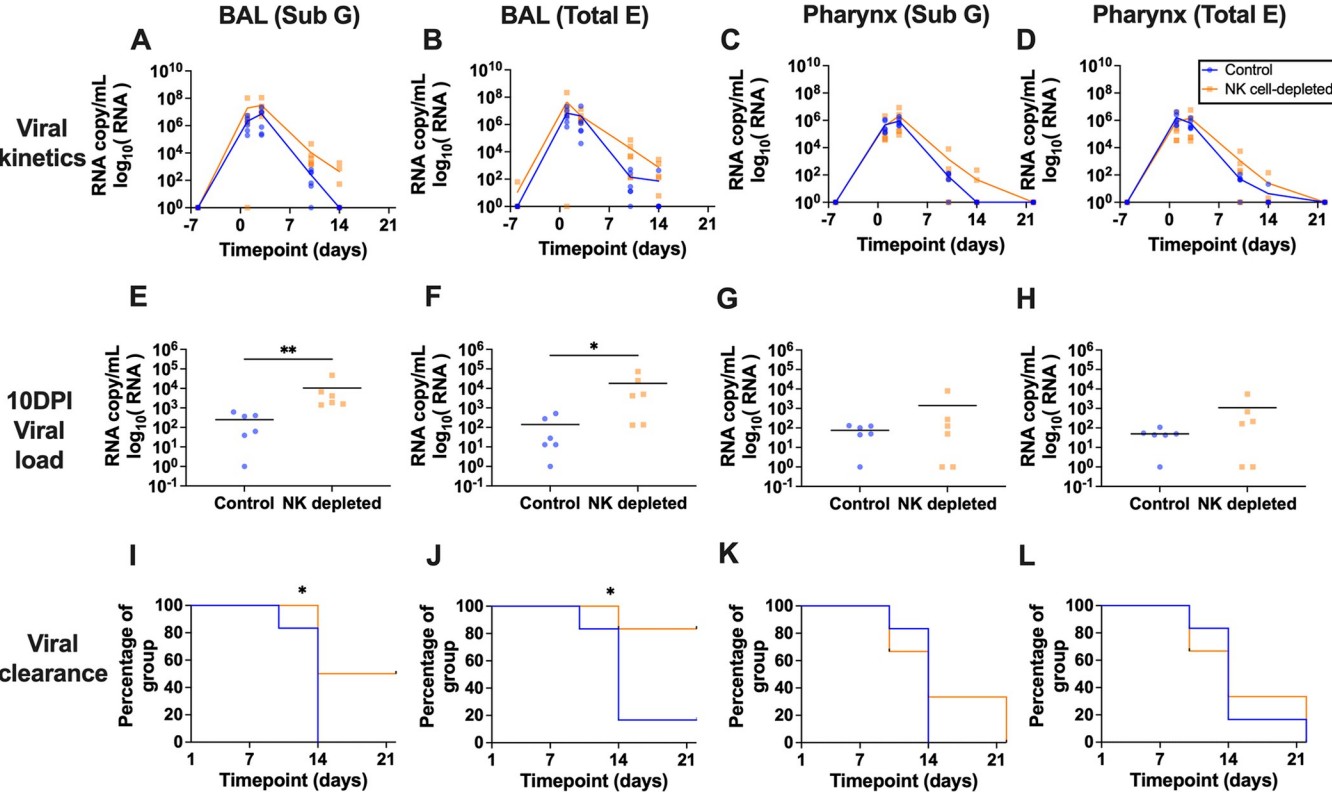

**Fig 2. Viral load analysis. (A)–(D)** Viral kinetics in BAL (Sub G), BAL (Total E), Pharynx (Sub G), and Pharynx (Total E); **(E)–(H)** Day 10 viral load in Pharynx (Sub G), Pharynx (Total E), BAL (Sub G) and BAL and **(I)–(L)** Viral clearance in BAL (Sub G), BAL (Total E), Pharynx (Sub G), and Pharynx (Total E). Black line indicates the mean. **E–H**—Mann Whitney test and **I–L**—Log-rank (Mantel-Cox) test * $p$-value ≤ 0.05 and ** $p$-value ≤ 0.01. Sub G–Subgenomic, Total E–Total envelope.

value = 0.024) in the NK cell-depleted group (mean 18063, range: 131–73713 RNA copies/mL) than in the control group (mean 140, range: 0–512 RNA copies/mL) (Fig 2F), a striking 2.5 log difference in the absence of NK cells. This trend was also true in the pharynx, albeit not significant (Fig 2G and 2H). Overall, these data clearly indicated that NK cell depletion resulted in increased virus replication. The duration of viral shedding in the control group was also shorter than in the NK cell-depleted group, with significant differences in the clearance curves of both VL measurements in the BAL compartment (p-value Sub G—0.0426 and Total E—0.025) (Fig 2I–2L). At 14DPI, none of the control group had detectable Sub G VL measurements in the BAL fluid and pharynx compartments, while 50% and 33% of the NK cell-depleted group, respectively, remained viremic. Similarly, as measured by total E VL, only 17% of the control group were viremic at 14DPI, while 33% and 83% of NK cell-depleted group had detectable virus in the BAL and pharynx, respectively.

## NK cell depletion exacerbates inflammatory cytokine profiles induced by infection

Next, we sought to measure soluble cytokine levels in both plasma and BAL samples. Not surprisingly, inflammation was robust during acute infection (3DPI) in all infected animals, and this was evident even in plasma (Fig 3A). Although some NK cell-depleted animals showed higher inflammatory mediators than controls, these differences were not statistically significant. However, in BAL inflammatory profiles (Fig 3B) were highly exacerbated by infection in all animals and further increased due to NK cell depletion. Of note, at 3DPI, interferon-alpha (IFNα) concentrations were significantly higher among the NK cell-depleted group (mean 1138.37pg/mL; range: 217.69–2308.83pg/mL) compared to the control group (mean 251.36pg/mL; range: 22.92–650.57pg/mL) (adjusted p-value = 0.0001) (Fig 3C).

Network analysis of plasma cytokines (S4 Fig) in the control group at 3DPI indicated that the cytokine-cytokine receptor interaction node involves multiple cytokines while at 14DPI, the complexity of this network decreases. In the control group at 3DPI, there were also many cytokines forming a viral protein interaction with cytokine and cytokine receptor node characterizing acute infection, which was not observed at 14DPI. The number of interacting

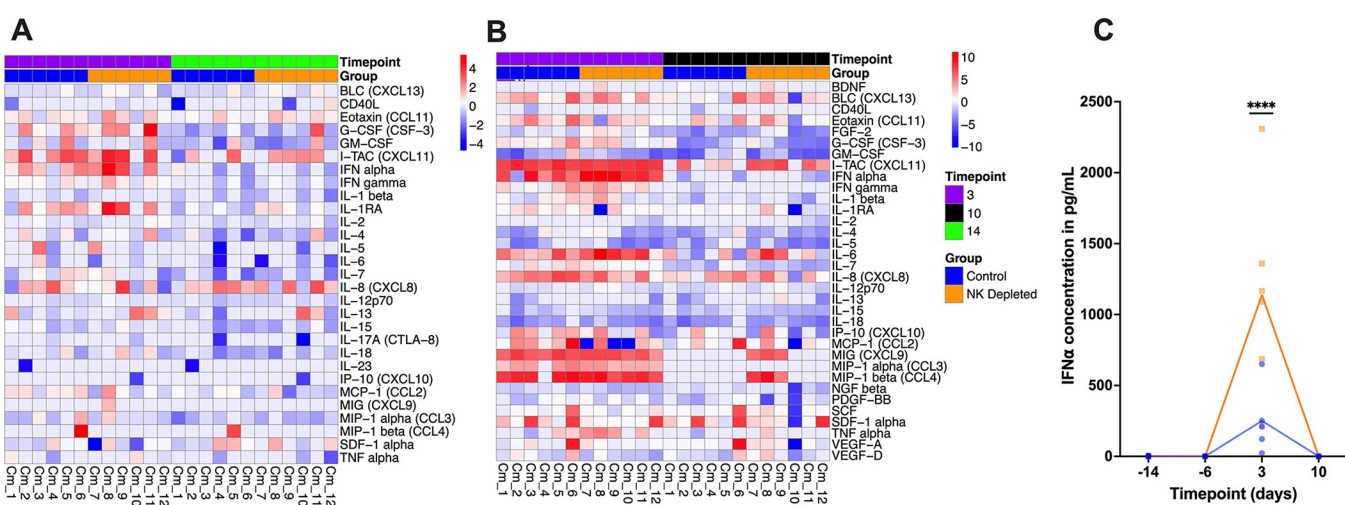

**Fig 3. Longitudinal profiles of cytokine level changes in both groups.** (**A**) Heatmap of the log2 fold change in cytokines from baseline to 3DPI and 14DPI observed in plasma; (**B**) Heatmap of the log2 fold change in cytokines from baseline to 3DPI and 10DPI observed in BAL; and (**C**) IFNα concentrations in BAL. **C**—Significance evaluated by Mixed effects analysis; **** adjusted p-value ≤ 0.001.

cytokines in the NK cell-depleted group was fewer, leading to a simpler network at both time-points. In the BAL cytokine network analysis (S5 Fig), cytokine-cytokine receptor interaction and viral protein interaction with cytokine and cytokine receptors nodes were highly complex during 3DPI in both groups but was observed at 10DPI only in the NK cell-depleted group indicative of their longer duration of viral shedding. Notably, at 10DPI in the NK cell-depleted group, the chemokines CCL11, CXCL13, CXCL8, and the cytokine IL-6 were upregulated. These findings potentially point to a critical role of NK cells in regulating immune recruitment and inflammation. Overall, these data confirm that SARS-CoV-2 infection induces significant inflammation in the lungs, and this is grossly increased in the absence of NK cells.

## NK cell ablation results in increased disease pathology during SARS-CoV-2 infection

Next, we wanted to assess if the increased virus replication and dissemination in the absence of NK cells had clear clinical consequences. We first evaluated general clinical and physiologic measurements. Liver enzymes (ALT, AST), renal metabolites (BUN, creatine) and reticulocyte counts were all significantly impacted by SARS-CoV-2 infection, with clear liver and kidney damage early in infection (Fig 4A–4F). Moderate changes in white blood cell, red blood cell, and lymphocyte counts, as well as body temperature and weight showed modest but non-significant fluctuations (S6 Fig). No differences in any of these measurements were found between NK cell-depleted and controls. It is also important to note that prior to infection no changes in any of these values were observed, indicative of the well-tolerated and non-toxic nature of the NK cell depletion strategy.

Next gross lung pathology for both groups was assessed using a standardized scoring system for SARS-CoV-2 pathology [57]. Although significant lung damage was found in all animals, the NK cell-depleted group had significantly higher lung lobe damage scores (mean 6.52; range: 0–15) compared to the control group (mean 2.86; range: 0–14). (p-value < 0.0001)

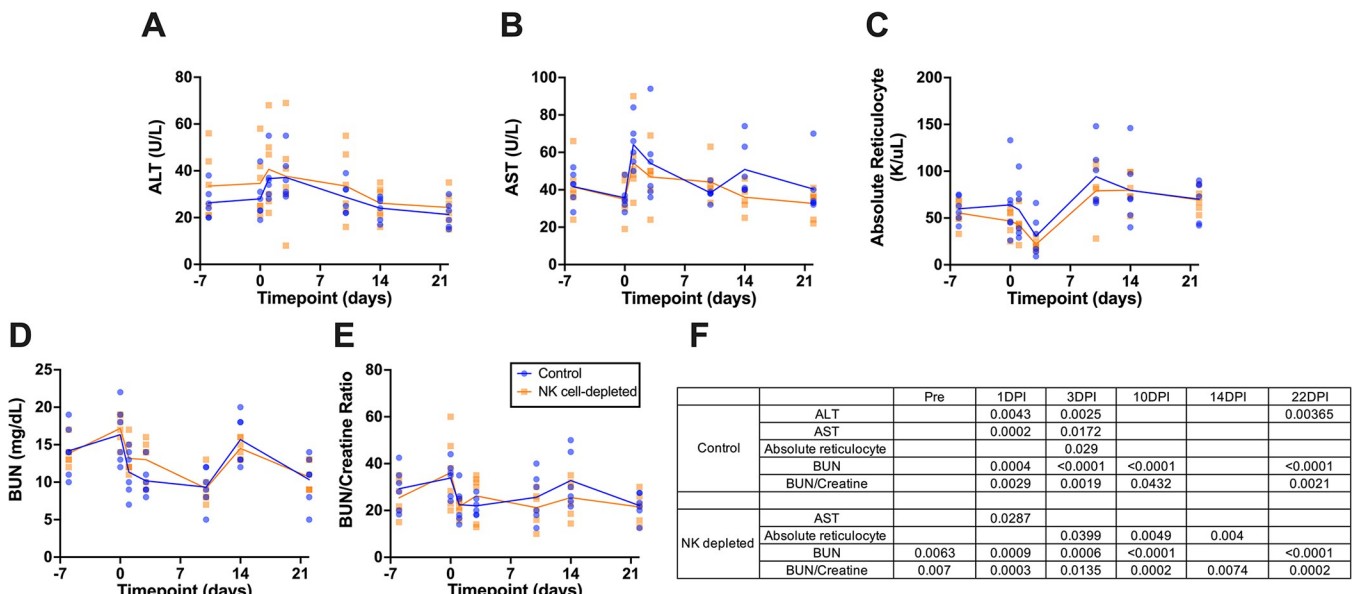

**Fig 4. Serum chemistry.** (A)–(E) Serum Chemistry measurements for ALT, AST, Absolute reticulocytes, BUN and BUN/Creatine ratio; (F) Table containing the p-value of each timepoint compared to time of infection (0DPI); F–Significance assessed by Two-way ANOVA; ALT—alanine transaminase, AST—aspartate aminotransferase and BUN—blood urea nitrogen.

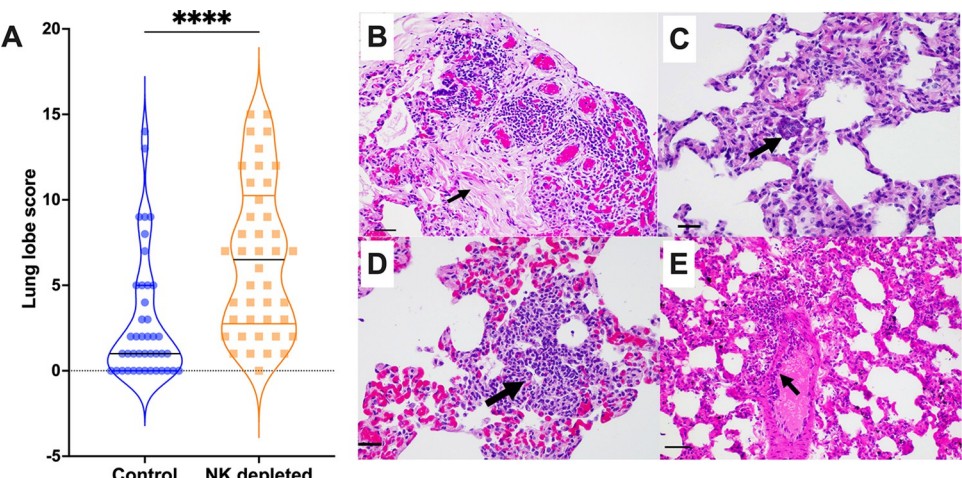

**Fig 5. Lung Pathology at necropsy. (A)** Lung lobe score; **(B)—(E)** H & E Staining on SARS-CoV-2 infected lungs of the NK cell-depleted group. Arrows indicate focal fibrosis, syncytia, type II pneumocyte hyperplasia, and endothelialitis respectively. Black line indicates the median and the lines in self color indicate quartiles. **B–E:** Magnification: 200x; Bar - 50uM. **A**–Significance assessed by Two-way ANOVA; **** $p$-value $< 0.0001$.

(**Fig 5A**). These results were particularly striking given the normally relatively low tissue damage found in NHP SARS-CoV-2 model. With closer examination by hematoxylin and eosin staining, greater lung damage in the NK cell-depleted group was clearly manifested by more frequent presence of focal fibrosis, syncytia, and pneumocyte hyperplasia compared to the control group (**Figs 5B–5D and S7A–S7C**). Interestingly two out of the six CM in the NK cell depleted group showed signs of endothelialitis, which was absent in all the six control CM (**Figs 5E and S7D**).

## Discussion

NHP models are pertinent to infectious disease research due to their physiological, genetic, and immunological similarities to humans [58]. The angiotensin-converting enzyme 2 (ACE2) receptors binding to SARS-CoV-2 is identical between NHP and humans whereas mouse ACE2 binding is limited [59,60]. We acknowledge that the NHP model in our study demonstrates transient infection and not the acute fulminant pathology of severe COVID-19 in humans and fails to address disease features like ARDS, coagulopathy, systemic sequelae, and mortality [61]. However, since SARS-CoV-2 infected NHP develop mild to moderate respiratory disease followed by full recovery, our model recapitulates the infection in humans [62]. The Delta variant was selected in this study due to its phenotypic advantages, including early infection kinetics and enhanced induction of inflammatory biomarkers [63]. Additionally, the Delta variant has a higher capacity to induce severe symptoms and an increased mortality rate in humans [64,65]. Airborne SARS-CoV-2 infection of CM has been reported to induce a model of COVID-19 more representative of human disease [66].

A previous SARS-CoV-2 infection study (Washington strain 2019-nCoV/USA-WA1/2020) on RM using an anti-CD8α depletion strategy concluded that T cells are not critical for recovery from acute infection while B cells and antibodies play a significant role in the recovery process and immune memory [32]. A CD8+ T cell depletion study using the same strain of virus as above indicated that CD8+ T cells may also contribute to protection when neutralizing antibody titers decline in an NHP model [67]. Proteomic and metabolomic analyses of SARS-CoV-2 infected (BetaCoV/Beijing/IME-BJ05/2020) CM have identified that the innate

immune system, neutrophil and platelet activation, and degranulation play central roles in mimicking a moderate COVID-19 disease [68]. These findings highlight the importance of other crucial immune system components in controlling acute SARS-CoV-2. NK cells are critical effector cells that modulate antiviral immunity prior to mounting of adaptive responses. The potential duality of NK cells in SARS-CoV-2 pathogenesis, i.e., whether they are beneficial or detrimental, is an important concept that has remained incompletely understood. Herein, we took an agnostic approach to this phenomenon using a well-established NK-cell depletion technique [23,48] in a cohort of CM to remove NK cells and subsequently challenge with the pathogenic SARS-CoV-2 Delta variant B.1.617.2.

In broad strokes, COVID-19 severity has been associated with decreasing NK cell numbers [69–71]. In keeping with these findings, our control group showed expanded NK cell numbers associated with virus control and peak expansion occurring at approximately 10DPI, consistent with previous findings [71]. We observed that at 3DPI, NK cells in the control group reached peak CD69 expression (**Fig 1D**). Upregulation of CD69 expression is an early signature of activation on lymphocytes [50,72,73], a regulator of cytokine, release, homing, and importantly recruitment to sites of inflammation [50,74,75]. Beyond these findings, NK cells in the control group also increased expression of the proliferation marker, Ki67 (**Fig 1F**), and expanded their functional capacity associated with virus replication (**Fig 1G**). Collectively, these data indicate a successful NK cell response to SARS-CoV-2 infection likely requires robust functionality, tissue homing, and proliferative expansion of the NK cell compartment.

The two most important pieces of data demonstrating a role for NK cells in SARS-CoV-2 come from the comparison of control and NK cell-depleted animals where (1) VL magnitude and duration increased in NK cell-depleted animals, and (2) lack of NK cells results in severe tissue pathology in all compartments studied. The systemic- and tissue-level NK cell depletion achieved by the anti-IL-15 mAb administration was highly successful (**Fig 1B**) with little to no adverse or off-target impacts, similar to previous reports by our group and others [23,48] and providing a clean model to evaluate the impacts on virology and pathology. The VL in our study in the pharynx peaked at 3DPI (**Fig 2**), consistent with reports by other groups using RM SARS-CoV-2 models that peak viral load occurs between 1-3DPI [76,77]. Although peak VL were only marginally higher, at all subsequent timepoints post infection, the NK cell-depleted group had higher VL compared to the control group, reaching significance at 10DPI. Concurrent to peak VL at 3DPI, we notice that the expression of activation marker CD69 on NK cells in the control group increased, followed by an increase in proliferation marker Ki67 expression at 10DPI, which is when peak NK cell frequencies in this model are observed. Notably, all animal tissue compartments in the control group cleared infection significantly faster than the NK cell-depleted group. Another phenotype of SARS-CoV-2-infected animals lacking NK cells was an overt increase in lung pathology and disease (**Fig 5**). The increased pathology was marked by focal fibrosis, syncytia, endothelialitis, and pneumocyte hyperplasia, all of which are reported by other groups in rodent and NHP models as well as fatal COVID-19 autopsy samples from humans [66,76,78,79]. Metadata analysis has indicated that post-COVID-19 pulmonary fibrosis manifested by the architectural distortion of the lung parenchyma affects 45% of recoverees [80]. One of the hallmarks of the Delta variant of SARS-CoV-2 is its enhanced ability to induce syncytia formation, which facilitates cell-to-cell viral spreading [63]. This phenomenon was observed in both groups in our study. Pulmonary edema and the formation of bronchiolar epithelial syncytial cells were also reported in a RM infection (SARS-CoV-2 USA-WA1/2020 strain) model but were far less severe than seen in our NK cell-depleted animals [76]. Remarkably the presence of endothelialitis, a characteristic feature of acute SARS-CoV-2 infection (observed 2–4 days post-infection in RMs) was observed in two out of the six CM in the NK cell-depleted group at necropsy and completely absent in the

control group [76]. Indeed, the fact that these animals demonstrate severe pathology as far out as 22DPI is in itself striking and our data suggest the lack of NK cells may be related to the more pathogenic disease normally seen in humans, aligning well with studies referenced above linking lower NK cells with clinical disease severity. Overall, these results indicate NK cells are critical to suppressing virus replication and associated severe lung disease.

One likely mechanism of increased disease pathogenesis was in the absence of NK cells increasing virus replication leads to over-production and activation of cytokine and chemo-kine networks, resulting in tissue damage. This level of excess inflammation is generally accepted as part of COVID-19 disease [81] and this provides a specific link to NK cell biology. As shown in **Fig 3**, NK cell-depleted animals had higher levels of inflammatory analytes in plasma, albeit only marginally. Interestingly, in the NK cell-depleted group, the concentration of peripheral stromal cell-derived factor-1 alpha (SDF-1α) increased post-infection. Research-ers have previously identified SDF-1α as a chemoattractant to recruit CD8$^+$ T and NK cells [82–85]. More importantly, increased inflammation in BAL was very striking in both control and NK cell-depleted animals. While the pro-inflammatory cytokine IFNα increased in both groups, levels in the NK cell-depleted group were five higher than the control group. IFNα lev-els have been reported to enhance the expression of ACE2 receptors on type II pneumocytes, resulting in increased SARS-CoV-2 infection and lung damage [86]. Excessive IFNα produc-tion also aggravates lung injury during influenza virus and severe acute respiratory syndrome–coronavirus 1 (SARS-CoV) infections in mouse models [87–89]. Upregulation of type 1 IFN pathway has been previously reported in transcriptomic analysis of RM infected with SARS--CoV-2 [90].

In conclusion, we report an activated, dynamically trafficking, and proliferating subset of functionally efficient circulating NK cells in acute SARS-CoV-2 infection. These NK cells are associated with lower VLs and shorter duration of viral shedding during acute disease. Further solidifying a critical role for NK cells, the experimental depletion of NK cells leads to signifi-cant increases in VLs, proinflammatory cytokines, and pathology in the lung. We surmise that NK cells play a crucial role in controlling SARS-CoV-2 pathogenesis by dampening replication and over immune activation in the lungs. Collectively, this study supports further investigation into NK cell-based vaccines and therapies for Coronaviruses among other infectious diseases.

## Methods

### Ethics statement

Animals were housed at Bioqual Inc. (Rockville, MD) and studies were carried out in accor-dance with the recommendations of the Guide for the Care and Use of Laboratory Animals of the National Institutes of Health with recommendations of the Weatherall report; "The use of non-human primates in research". All blood and tissue samplings were collected as part of study protocol #21–071. Protocol #27–071 was reviewed and approved by the Bioqual Institu-tional Animal Care and Use Committee. The diet of the animals included standard monkey chow diet supplemented daily with fruit and vegetables and water ad libitum. Social enrichment was provided to the animals and was overseen by veterinary staff. Animal health was monitored daily and if any signs of significant weight loss, disease or distress, they were evaluated clinically and then provided dietary supplementation, analgesics and/or therapeutics as necessary.

### Animal study design and sampling

Twelve experimentally naïve, age-matched CM (*Macaca fasicularis)* of Cambodian origin were selected for this study. All animals were housed at BIOQUAL, Inc. (Rockville, MD, USA) and were free of simian retrovirus type D and simian T-lymphotropic virus type 1.

Half of the cohort received weight-dependent anti-IL-15 (NIH funded NHP Reagent Resource NIAID U24 AI126683, RRID: AB_2716329) infusions three times during the duration of the study to achieve NK cell depletion and are referred to as the NK cell-depleted group (**Fig 1A**). The first two infusions were administered at 20mg/kg dosage two weeks prior to challenge and one day after challenge, while a third dose of 10mg/kg was administered one-week post-challenge. The remaining six CM received 4mL of phosphate-buffered saline (PBS) placebo corresponding to the time of infusion. All animals were challenged via intranasal and intratracheal routes with SARS-CoV-2-hCOV-19/USA/MD-HP05647/2021 (B.1.617.2) Delta Variant (BEI NR-56116 Lot #: 70047614) [91].

Throughout the duration of the study, animals were clinically evaluated for responsiveness, body weight, respiratory rate and effort, fecal consistency, and body temperature. The examinations also involved monitoring of discharge, body condition scores, and hydration scores while sedated and awake. Serum chemistry measurements included concentration of white blood cells (WBC), red blood cells (RBC), hemoglobin (HGB), hematocrit (HCT), mean corpuscular volume (MCV), mean corpuscular hemoglobin (MCH), mean corpuscular hemoglobin concentration (MCHC), alkaline phosphatase (ALP), alanine transaminase (ALT), aspartate aminotransferase (AST), creatine kinase, gamma-glutamyl transpeptidase (GGT), albumin, total protein, globulin, total bilirubin, blood urea nitrogen (BUN), creatinine, cholesterol, glucose, calcium, phosphorous, chloride, potassium and sodium, ratios of BUN/creatine and sodium/potassium and neutrophil, eosinophil, lymphocytes, monocytes, reticulocytes concentrations as well as percentages.

## Virologic measurements

SARS-CoV-2 Total envelope (E) and nucleocapsid (N) gene subgenomic mRNA were measured by one-step RT–qPCR [92,93]. E and N genes were cloned into pCDNA3.1 and used as *in vitro* transcription templates. Transcribed RNA was generated with the MEGAscript T7 Transcription Kit (ThermoFisher) and purified with MEGAclear Transcription Clean-Up Kit (ThermoFisher). The obtained pure RNA was quantified and used as standards for qPCR. RNA extracted from NHP samples was quantified using TaqMan Fast Virus 1-Step Master Mix (ThermoFisher) and custom primers and probes targeting the E gene sgRNA (forward primer, 5′-CGATCTCTTGTAGATCTGTTCTCE-3′; reverse primer, 5′-ATATTGCAGCAG-TACGCACACA-3′; probe, 5′-FAM-ACACTAGCCATCCTTACTGCGCTTCG-BHQ1-3′) or the N gene sgRNA (forward primer, 5′-CGATCTCTTGTAGATCTGTTCTC-3′; reverse primer, 5′-GGTGAACCAAGACGCAGTAT-3′; probe, 5′-FAM-TAACCAGAATGGAGA ACGCAGTG GG-BHQ1-3′). A QuantStudio 3 Real-Time PCR System (Applied Biosystems) or a StepOnePlus Real-Time PCR System (Applied Biosystems) was used for real-time PCR reactions. Standard curves were used to calculate E or N sgRNA in copies/ml. The limit of detection for both E and N sgRNA assays were 12.5 copies/reaction or 150 copies/ml of BAL, nasal swab, or nasal wash. The total E measurements were included for SARS-CoV-2 infection monitoring, while the subgenomic N gene mRNA is a potential measure of replicating virus and is used to differentiate productive infection from input virus [94].

## Mononuclear immune cell isolation

Processing of blood and tissue samples for cell collection was carried out using protocols previously optimized in our laboratory [23,95,96]. Blood samples were collected in an anticoagulant (ethylenediaminetetraacetic - EDTA)-coated tube. To isolate peripheral blood mononuclear cells (PBMC), the blood cellular components was separated from the plasma using centrifugation. Plasma was collected and stored for cytokine evaluation. The cellular component of the

blood was resuspended in Dulbecco's phosphate-buffered saline (1X DPBS, Life technologies, catalog number: 14190144) and subjected to standard density gradient (Ficoll-Plaque, VWR International LLC, catalog number: 95038–168) centrifugation. The PBMCs at the interface of the Ficoll and DPBS layer were collected and contaminating red blood cells were lysed using a hypotonic ammonium chloride-potassium chloride solution. The cells were washed and counted before subsequent analyses.

For LN and spleen tissues, excess fat was trimmed from the tissues and then mechanical disaggregation of the tissues was performed using an Octodissociator. The cell solutions were passed through a 70μm filter to remove any debris. The CR samples were subjected to mechanical dissociation in the Octodissociator, along with enzymatic digestion using collagenase before collecting, filtering, and washing the cells. As with the blood samples, the cells from these tissues were collected, washed, and counted before subsequent analyses.

## Flow cytometry

For whole blood staining, 100μL of blood was aliquoted and an antibody cocktail (**S1 Table**) was added for cell staining. RBCs were lysed then cells were washed and fixed in 2% Formaldehyde (Fisher Scientific, Catalogue number:NC9200219). For the cells isolated from tissues, $2x10^6$ cells were aliquoted, washed, and incubated in a diluted (1,1000 in 1X DPBS) near infrared (NIR) live/dead stain (**S1 Table**). Cells were then washed and fixed before flow cytometry analysis with a BD FACSymphony A5 Cell Analyzer. A minimum of 300,000 events were acquired and the downstream analyses were conducted using FlowJo (version 10.8.1, BD Life Sciences). The gating strategy used is shown in **S8A Fig**. In brief, the cells were gated on lymphocytes, followed by single cells and live CD45$^+$ cells. The T cells were gated based on CD3 expression followed by CD4 and CD8 expressions. Based on the expression of CD62L and CD95, the T cells were subset into naïve, effector memory, and central memory populations. From the CD3$^-$ population, CD14 and CD20 negative population was selected and gated on NKG2A/C positivity as NK cells. This is considered the standard phenotype for analysis of bulk macaque NK cells [56,97]. CD16, CD56, and CD69 expression on this population was analyzed.

## Intracellular cytokine staining

Multi-parameter intracellular cytokine staining assays was performed to evaluate the functionality of NK cells. A degranulation mixture containing 5μL/test CD107a (**S1 Table**), 1μL/test of Brefeldin (BD Biosciences, catalog number: 555029) and 0.7μL/test Monensin (BD Biosciences, catalog number: 554724) was added to all samples. Cell stimulation cocktail (eBioscience Cell Stimulation Cocktail (500X), Thermo Fisher Scientific, catalogue number: 00-4970-93) containing phorbol 12-myristate 13-acetate (PMA) and ionomycin was added to the stimulated wells. Unstimulated cells were used as negative controls. Following 6 hour incubation cells were stained with near-IR live/dead dye, washed, then stained with surface antibody mixture (**S1 Table**). Cells were then washed and incubated with 100μl of Fixation medium A (Life technologies, catalogue number: GAS001S100), followed by staining with intracellular antibody mixture diluted in Permeabilization medium B (Life technologies, catalogue number: GAS002S100). Fixed cells were then analyzed by flow cytometry on a BD FACSymphony A5 Cell Analyzer. A minimum of 300,000 events were acquired and the downstream analysis was conducted using FlowJo (version 10.8.1). The gating strategy is shown in **S8B Fig**. Additionally, the expression of MIP-1β, IFN-γ, TNF-α, and CD107a on the NK cells were analyzed.

## Luminex

Banked plasma and BAL samples were deactivated for viral contaminants by diluting 50/50 with 2% triton. For generation of standard curves, manufacturer-provided lyophilized standards were reconstituted and prepared by 4-fold serial dilutions for a total of eight standards as described in the kit insert. These standards were analyzed in duplicate on a 96-well optical plate alongside the prepared plasma samples, which were also run in duplicate. Two additional wells were run without standards or samples to provide background measurements. The concentrations of cytokines and chemokines were measured via Luminex xMAP technology utilizing the Procartaplex Human Cytokine/Chemokine Convenience Panel 1A 34-plex (ThermoFisher EPXR340-12167-901) in accordance with the manufacturer's instructions. Analysis of the plate wells was performed using the Luminex 200 (Luminex Corporation) which was calibrated, and performance was validated as per instrument validation protocol. Measurements were reported with the xPONENT 4.2 software (Luminex Corporation).

## Pathology

Lungs harvested at 22DPI were evaluated utilizing a histopathology technique adapted from a previously published study [57]. The tissue was fixed in 10% formalin and blocks were sectioned at 5μm thickness onto slides. The slides were incubated for 30–60 minutes at 65˚C and then deparaffinized in xylene. The tissue was rehydrated through a series of graded ethanol to distilled water. Sections were stained with haematoxylin and eosin. Blinded evaluation and scoring were performed by a board-certified veterinary pathologist (A.J.M.). For each CM, four representative samples from right and three samples from left lungs were evaluated and were scored independently. The scoring was performed based on the identification of the following lesion features: interstitial inflammation and septal thickening, interstitial infiltrate (eosinophils), interstitial infiltrate (neutrophils), hyaline membranes, interstitial fibrosis, alveolar infiltrate (macrophages), bronchoalveolar infiltrate (neutrophils), epithelial syncytia, type II pneumocyte hyperplasia, bronchi infiltrate (macrophages), bronchi infiltrate (neutrophils), bronchi (hyperplasia of bronchus-associated lymphoid tissue), bronchiolar or peribronchiolar infiltrate (mononuclear cells), perivascular infiltrate (mononuclear cells), and endothelialitis. Each feature assessed was assigned a score of 0, no substantial findings; 1, minimal; 2, mild; 3, moderate; 4, moderate to severe; 5, marked or severe. Scores were added for all lesions across all lung lobes for each CM, for a maximum possible score of 600 for each animal.

## Statistical analysis

Statistical analyses were conducted using GraphPad Prism (v9, GraphPad Software, USA) unless specified otherwise. Descriptive statistics of data arrays were summarized with measures of central tendency (mean or median) or dispersion (standard deviation or interquartile range) depending on normality of distributions. The analysis of each cell subset was performed utilizing its percentage based off the total number of live CD45$^+$ lymphocytes. Statistically significant associations were explored with appropriate parametric or non-parametric tests. Multiple Mann-Whitney $U$ to compare longitudinally between groups across different cell subsets, Two-way ANOVA to compare the cell subsets across visits, One-way ANOVA was used to evaluate longitudinal trends in NK cell subsets in the control group, Mann-Whitney $U$ test was used to compare NK cell levels between the groups in the tissue and viral loads, Log-rank (Mantel-Cox) test was used to evaluate differences in the VL clearance curves and Wilcoxon matched-pairs signed rank test to compare trafficking markers between pre timepoint and necropsy in the control group in tissue).

Heatmap figures for cytokine levels in plasma and BAL samples were generated in R using the *pheatmap* package [98]. Briefly, data were loaded into R and log2 fold changes were calculated by comparing indicated timepoints versus pre-infection timepoints. Fold changes were then log2 transformed and heatmaps generated without row or column scaling.

For the cytokine networking analysis, Luminex cytokine data were loaded into R v4.3. Samples were grouped by study group and timepoint, and means for each cytokine were taken. Fold change values were then calculated for each group and timepoint by dividing each timepoint by the pre-infection timepoint and then log2 was transformed. Cytokine names, log2 fold change values, and *p*-values were assembled into a data frame and as input for the standard pathfinder [99] workflow utilizing the KEGG database for interaction network analysis. Plots were generated with the term_gene_graph() function run on the result of pathfindR analysis.

## Supporting information

**S1 Fig. Longitudinal observation of impact of anti-IL15 treatment on lymphocytes and T cell subsets. (A)** Percent total CD4$^+$ T cells; **(B)** Percent total CD8$^+$ T cells; **(C)** Percent total B cells; **(D)** Percent naïve CD4$^+$ T cells of total CD4$^+$ T cells; **(E)** Percent central memory CD4$^+$ T cells of total CD4$^+$ T cells; **(F)** Percent effector memory CD4$^+$ T cells of total CD4$^+$ T cells; **(G)** Percent naïve CD8$^+$ T cells of total CD8$^+$ T cells; **(H)** Percent central memory CD8$^+$ T cells of total CD8$^+$ T cells; **(I)** Percent effector memory CD8$^+$ T cells of total CD8$^+$ T cells; Significance assessed by Two-way ANOVA.
(TIFF)

**S2 Fig. Percentage of NK cells in all compartments at the terminal timepoint stratified by group. Blue**–control group and **Orange**–NK cell-depleted. Black lines indicate mean and the self-colored lines indicate standard deviation. AX LN–Axillary lymph node; ING LN–Inguinal lymph node; Med LN–Mediastinal lymph node; Mes LN–Mesenteric lymph node; CR–Colorectal biopsy. Significance assessed by Multiple Mann Whitney tests; * adjusted *p*-value ≤ 0.05
(TIFF)

**S3 Fig. Longitudinal NK cell subset distribution in blood over time in the control group.** Table below main figure indicates the statistically significant differences in frequency from 10DPI to 22DPI. Significance assessed by One-Way ANOVA (* adjusted *p*-value ≤ 0.05; *** adjusted *p*-value ≤ 0.001).
(TIFF)

**S4 Fig. Term enrichment analysis performed on log2(Fold Change) with pathfinder in plasma Luminex assay. (A)**–**(B)** Control group 3DPI and 14DPI; **(C)**–**(D)** NK cell-depleted group 3DPI and 14DPI.
(TIFF)

**S5 Fig. Term enrichment analysis performed on log2(Fold Change) with pathfinder in BAL. (A)**–**(B)** Control group 3DPI and 10DPI **(C)**–**(D)** NK cell-depleted group 3DPI and 10DPI.
(TIFF)

**S6 Fig. Longitudinal serum chemistry and physical parameter observations of all cynomolgus macaques. (A)** White blood cells; **(B)** Red blood cells; **(C)** Hemoglobin; **(D)** Percentage of lymphocytes; **(E)** Fold change in body weights and **(F)** Fold change in rectal temperatures. K–thousand; M- million; g- gram. Significance assessed by Two-way ANOVA.
(TIFF)

**S7 Fig. Lung Pathology at necropsy. (A)—(D)** H & E Staining on SARS-CoV-2 infected lungs of the control group indicating focal fibrosis, syncytia, type II pneumocyte hyperplasia, and endothelialitis respectively.
(TIFF)

**S8 Fig. Flow cytometry gating strategies for (A)** NK cell and T cell phenotyping; **(B)** Intracellular cytokine staining.
(TIFF)

**S1 Table. List of antibodies used.**
(DOCX)

## Author Contributions

**Conceptualization:** Stephanie Jost, R. Keith Reeves.

**Data curation:** Kyle Kroll, R. Keith Reeves.

**Formal analysis:** Harikrishnan Balachandran, Kyle Kroll, Karen Terry, Cordelia Manickam, Rhianna Jones, Amanda J. Martinot, Ankur Sharma, Stephanie Jost.

**Funding acquisition:** Stephanie Jost, R. Keith Reeves.

**Investigation:** Harikrishnan Balachandran, Tammy Hayes, Amanda J. Martinot.

**Methodology:** Tammy Hayes, Amanda J. Martinot, Ankur Sharma.

**Project administration:** Rhianna Jones, Griffin Woolley, Tammy Hayes, Ankur Sharma, Mark Lewis, R. Keith Reeves.

**Resources:** Mark Lewis.

**Supervision:** R. Keith Reeves.

**Writing – original draft:** Harikrishnan Balachandran, Kyle Kroll, Karen Terry, Cordelia Manickam, Rhianna Jones, Griffin Woolley, Tammy Hayes, Amanda J. Martinot, Ankur Sharma, Mark Lewis, Stephanie Jost, R. Keith Reeves.

**Writing – review & editing:** Harikrishnan Balachandran, Kyle Kroll, Karen Terry, Cordelia Manickam, Rhianna Jones, Griffin Woolley, Tammy Hayes, Amanda J. Martinot, Ankur Sharma, Mark Lewis, Stephanie Jost, R. Keith Reeves.

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
