## [Decision Letter · Decision Letter 0]

21 May 2024

Dear Dr. Reeves,

Thank you very much for submitting your manuscript "NK cells are critical for in vivo control of SARS-CoV-2 replication and suppression of lung damage" for consideration at PLOS Pathogens. As with all papers reviewed by the journal, your manuscript was reviewed by members of the editorial board and by several independent reviewers. The reviewers appreciated the attention to an important topic. Based on the reviews, we are likely to accept this manuscript for publication, providing that you modify the manuscript according to the review recommendations.

Sincerely,

Nichole R. Klatt

Guest Editor

PLOS Pathogens

Michael Letko

Section Editor

PLOS Pathogens

Michael Malim

Editor-in-Chief

PLOS Pathogens

orcid.org/0000-0002-7699-2064

Reviewer Comments (if any, and for reference):

Reviewer's Responses to Questions

**Part I - Summary**

Reviewer #1: While the contribution of T and B cells in SARS-CoV-2 infection have been largely characterized, few groups have investigated the effect of NK cells during infection. In human studies, hypothesized contributions of NK cells have ranged from being crucial early antiviral responders to mediators of hyperinflammation and lung damage in patients. Utilizing a well-established NK cell depletion strategy in cynomolgus macaques, Balachandran et al directly assess the contribution of NK cells on the clearance and pathology of SARS-CoV-2 infection in a NHP model.

In the control group, NK cells exhibited activation and proliferation markers (CD69 and Ki67) as well as functional toxicity and cytokine production potential as early as 3 DPI, suggesting a robust acute NK cell response. While peak viral loads were not different between the control and NK cell depleted group, NK cell depleted animals had higher viral loads at 10 DPI and exhibited prolonged viremia compared to control animals. Furthermore, NK cell depleted animals exhibited higher IFNa and TNFa concentrations at 3 DPI and exhibited higher lung tissue damage at necropsy post infection.

While this is a novel approach to directly assess the function of NK cells in SARS-CoV-2 infection, I have some concerns that their depletion strategy administered concurrently during acute SARS-CoV-2 infection may complicate some of their analysis presented. Furthermore, the statistical claims in the paper are lacking rigor, and results are overstated Some sections could also use some clarification and polishing. Furthermore, figures are missing formatting, error bars and P-values.

Reviewer #2: Balachandran and coworkers provide an interesting interventional experiment in a nonhuman primate model to evaluate the role of NK cells in SARS-CoV2 pathogenesis. They nicely summarize the literature in humans with COVID-19 that has suggested positive or negative roles that NK cells could be playing in disease, making this an excellent question to pose. Cynomolgus macaques, which have been validated as a model for SARS-CoV2 infection, were inoculated via intranasal and intratracheal routes with Delta variant B.1.617.2 either with or without NK depletion by an anti-IL15 antibody. This group has published experience with this protocol in rhesus macaques undergoing SIV infection in which depletion of NK cells prior to challenge showed an enhancement of SIV replication acutely. In cyno macaques, the anti-IL15 antibody produced impressive effects in depleting NK cells for the 21-day duration of the post-infection follow-up, and no effect on CD8 T cells was observed. A comprehensive set of assays demonstrated: 1) control animals exhibited a rapid expansion of NK cells and transient expression of markers consistent NK activation, proliferation and possibly trafficking; and 2) NK cell depletion led to higher and more persisting levels of virus in BAL fluid and pharyngeal secretions; increased levels of pro-inflammatory cytokines in lungs; and more histological damage in lungs as assessed by blinded pathological review and scoring. The Reeves laboratory is among the leaders in the NK field and in developing NHP models to evaluate the role of NK cells in pathogenesis and vaccines. The study is well designed and rigorously conducted and provides new information that adds to our understanding of early immune responses during acute SARS-CoV2 infection. The finding that pathological outcomes and proinflammatory cytokine signaling pathways are enhanced following NK depletion is especially interesting in suggesting that this damage resulted from more persistent exposure to virus or dysregulatory effects on inflammatory responses.

Reviewer #3: In this manuscript by Balachandran et al, the authors use a cynomolgus macaque model of NK cell depletion (via sustained IL-15 depletion) prior to SARS-CoV-2 infection (delta variant) and find that viral titers are somewhat higher on day 10 post infection in depleted animals than they are in control animals. These data suggest an important role for NK cells in viral clearance. Additionally, the study provides a comprehensive assessment of NK responses to this virus in an important and translational model. The study is well done and the manuscript is well-written. There are caveats to the approach that could impact interpretation of the data that need to be addressed.

**Part II – Major Issues: Key Experiments Required for Acceptance**

Reviewer #1: Major Comment 1: Anti-IL15 mAb treatment may stimulate inflammatory responses, potentially complicating analysis on differences in cytokine levels during SARS-CoV-2 infection in NK cell depleted vs control groups.

a. Why was anti-IL15 administered concurrently during the SARS-CoV-2 challenge (1 and 7 DPI) when it could potentially complicate analysis? Reactions to anti-IL15 or homeostatic changes in other cell populations could further contribute to increased pro-inflammatory cytokine levels. Given the dosing strategy, it is unclear if the absence of NK cells during acute infection or the just the anti-IL15 administration itself could be causative of increased inflammatory cytokine levels observed.

b. It appears samples were collected at -6/7 DPI (post depletion but pre infection), were there comparisons done between depleted/control group in cytokine levels at this timepoint? If there wasn’t significant difference in cytokine profiles between depletion/control groups at this timepoint this could potentially validate the anti-IL15 dosage regime did not complicated analysis.

Major Comment 2: Statistical analysis of claims in paper are lacking rigor.

a. Statistical differences in viral loads are based on a selective analysis of a single time point (day 10), when differences in viral loads between the NK cell-depleted and the control groups were greatest (Fig. 2E, 2F, 2G & 2H).

b. Statistical differences in viral load clearance from BAL and pharynx are not believable. With six animals per group and a difference in viral clearance at a single time point, it is difficult to believe that differences in the rate of viral clearance all reach a significance of p<0.0001 (Fig. 2I, 2J, 2K & 2L). This is particularly true for Fig. 2L, where clearance rates for the NK cell-depleted and control groups nearly overlap. The raw data needs to be provided to independently verify these claims.

c. While the manuscript claims there was no difference in T cells (bulk and CD4/CD8) between the depleted and control group, no statistical analysis was done to validate that claim. There is no statistical analysis presented for Figure 1C, and S1AB. Furthermore, it is not presented if there was any difference in B-cell homeostasis between the two groups.

d. P-values and individual data points are missing from a number of the figures (Figures 1, 3, 4, 5, S1, S2, S3, S6). It appears some of these issues are fixed on the previous preprint in Research Square, but it is an oversight if these were not corrected in the submission.

Major Comment 3: Given the relatively minor differences in viral loads between the experimental and control groups and the selective statistical analysis of the viral load data, the title of the manuscript claiming that NK cells are critical for in vivo control of SARS-CoV-2 replication is grossly overstated and needs to be revised.

Reviewer #2: The only major point to note is that the authors should say more about what the NHP model(s) have been able to show for SARS-CoV2 infection and what their limitations have been. This model is excellent for demonstrating transient infection, but not for the recapitulating the acute fulminant pathology of severe COVID-19 in humans. Just a few comments for a general audience with references on the state-of-the art for this NHP model would provide a useful perspective while not mitigating their observations.

Reviewer #3: (No Response)

**Part III – Minor Issues: Editorial and Data Presentation Modifications**

Reviewer #1: Minor comment 1: The cytokine network analysis is somewhat lacking and poorly worded (line 225-237). Furthermore, no description of the methodology is included in methods section.

a. Does the analysis break down what networks are involved in “Viral protein interaction with cytokine and cytokine receptor”? I understand this term is enriched only at 3DPI for the control group whereas it was enriched at 3 DPI and 10 DPI for NK cell depleted, potentially consequence of the prolonged viremia, however, the explanation is poorly worded.

b. Furthermore, it is referenced that complexity of the networks differs between timepoints, but it is not explained what relevance or conclusions could be drawn from this difference.

Minor Comment 2: Minor grammar/wording comments outlined below:

a. Line 76 “… (for example, NKG2D and NKP80)” for example is redundant, can be deleted.

b. Line 74-79 is a run-on sentence and confusingly written. Please reword.

c. Line 96,97 “NK cells in humans decline to 55%...” Believe it should be “declined by 55%”?

d. Line 124 in section title – Should be SARS-CoV-2 instead of SARSCoV2s to be consistent.

e. Figure 1A references a -6 DPI timepoint when others reference -7 DPI. Please correct/confirm

f. Line 268 “our control showed expanded NK…” should be our control group showed.

g. Line 280 “… role for NK cells in SARS-CoV-2, come from…” comma is unnecessary.

Reviewer #2: 1. In the abstract and at several points in the paper the term “viremia” is used. Since SARS-CoV2 is being measured in sections and not blood (as was the case in their JVirol paper on the role of NK cells in SIV infection), this term should not be used. Shedding would seem to be more appropriate.

2. Line 160: describing human populations of NK cells in peripheral blood, they should refer to the “CD56brightCD16- cytokine-secreting…” cells.

3. Supp Fig 3 did not reproduce well in the version of the manuscript that came across on line. It looks like the margins of the bars were deleted. Only the CD56+CD16- and CD56-CD16- cell populations are shown in the bars.

4. Line 190: For the general audience, add a descriptor or a line in the methods to indicate what subgenomic (Sub G) and total envelope (Total E) are actually measuring.

5. In Figure 3, Panels C and D, some data may not have transferred. Controls are not shown and the “Bar” on Day 3 is undefined.

6. In Fig 5, the histology panels B, C, D and E show only NK-depleted samples. It seems odd not to have any pictures of the Controls. Perhaps add more pictures in a supplemental figure? Also, the differences noted in histology producing the “Lung Lobe Score” were determined (Line 470) in a blinded manner “by a board-certified pathologist (A.J.M).” It’s not clear from the author list who this person is. If this is Amanda Martinot, please include her middle initial in the author list. If she was the pathologist, it would be worth noting that she was also the pathologist who apparently did the scoring in the cited Nature 2021 manuscript (Ref 59).

Reviewer #3: Overall, the manuscript could benefit from a more thorough description of how this study fits in the plethora of NHP SARS-CoV-2 studies. This is not the first study to use cynomolgus macaques with this virus nor the first to use the delta variant but it appears no other studies with this species or use of this variant in any NHP model are discussed or cited.

The use of the delta variant is an interesting and possibly important choice. Do the authors think the use of this virus, which showed somewhat increased virulence in humans and animal models, impacted their results? If so, this should be discussed.

Importantly, a previous study was conducted with depletion of NK cells (as well as CD8 T cells) prior to infection using the anti CD8a antibody, and showed no effect on viral titer (Hasenkrug et al, DOI: 10.1128/mBio.01503-21). There were important differences in that study besides use of a different depletion technique (that study used rhesus macaques infected with the WA1 viral isolate). That manuscript should be cited and the different findings discussed.

The authors note that NK cell expansion peaked on day 10 in control animals, coincident with higher levels of viral RNA in the lung in depleted animals, but it appears many/most other markers of NK activity peak on day 3, coincident with the peak of viral titer in both groups of animals. This is worth mention.

Previous studies have shown that IL-15 depletion does transiently deplete effector memory CD4 and 8 T cells and the authors cite one of these manuscripts (citation #50), but it’s not clear if the Tem cells were measured in this study? On a similar note, by day 10 when viral differences are noted between groups, adaptive immune responses are likely ramping up but no measures of either T cell or humoral immunity are shown. Are these data available? Given IL-15 depletion has a known (though subtle) impact on T cell immunity, is it possible the viral titer differences on day 10 are due to differences in T cell immunity (or another non-NK effect)?

I believe the word “viremia” is defined as describing virus in blood. This word is used throughout the manuscript to describe viral titer in other compartments. This should be changed.

Supplemental Figure 1 suggests lower CD8 T cell numbers in NK depleted animals, particularly at later time points that are important for this study, but no statistics are shown. Is this difference significant?

Supplemental Figure 2 only shows black lines, not orange or blue as indicated in the legend.

PLOS authors have the option to publish the peer review history of their article (what does this mean?). If published, this will include your full peer review and any attached files.

Reviewer #1: No

Reviewer #2: No

Reviewer #3: No

Figure Files:

Data Requirements:

Reproducibility:

References:

---

## [Decision Letter · Decision Letter 1]

22 Jul 2024

Dear Dr. Reeves,

We are pleased to inform you that your manuscript 'NK cells modulate in vivo control of SARS-CoV-2 replication and suppression of lung damage' has been provisionally accepted for publication in PLOS Pathogens.

Best regards,

Nichole R. Klatt

Guest Editor

PLOS Pathogens

Michael Letko

Section Editor

PLOS Pathogens

Michael Malim

Editor-in-Chief

PLOS Pathogens

orcid.org/0000-0002-7699-2064

Please note one minor comment from reviewer 3 and add one sentence to the manuscript to address this.

Reviewer Comments (if any, and for reference):

Reviewer's Responses to Questions

**Part I - Summary**

Reviewer #2: The authors have responded clearly and adequately to the criticisms of all 3 reviewers.

Reviewer #3: (No Response)

**Part II – Major Issues: Key Experiments Required for Acceptance**

Reviewer #2: I have no major issues. All have been addressed in the thorough responses to my review and the other 2 reviewers.

Reviewer #3: (No Response)

**Part III – Minor Issues: Editorial and Data Presentation Modifications**

Reviewer #2: None

Reviewer #3: I thank the authors for their responses to my critiques. Further, I thank the authors for adding the discussion of the Hasenkrug et al manuscript (DOI: 10.1128/mBio.01503-21), but their added text is focused on the role of T cells, which wasn't the point of my comment. My suggestion to cite and discuss that manuscript was to point out that that study used an anti-CD8alpha antibody so would have depleted NK cells, perhaps even more potently than targeting of IL-15. Yet that study did not see an impact of depletion on SARS-CoV-2 replication or subsequent disease. thus their results do not support a role for NK cells in clearance of virus or protection from disease, which obviously stands in contrast to the present study. This needs to be mentioned and discussed. I do think there are enough differences between that study and the present one that it does not invalidate the present study, by any means. however, it's important that these differences be highlighted and discussed for the reader to assess for themselves.

PLOS authors have the option to publish the peer review history of their article (what does this mean?). If published, this will include your full peer review and any attached files.

Reviewer #2: No

Reviewer #3: No

---

## [Editor Report · Acceptance letter]

8 Aug 2024

Dear Dr. Reeves,

We are delighted to inform you that your manuscript, "NK cells modulate in vivo control of SARS-CoV-2 replication and suppression of lung damage," has been formally accepted for publication in PLOS Pathogens.

Best regards,

Michael Malim

Editor-in-Chief

PLOS Pathogens

orcid.org/0000-0002-7699-2064